# Effects of Extracellular Vesicles from Osteogenic Differentiated Human BMSCs on Osteogenic and Adipogenic Differentiation Capacity of Naïve Human BMSCs

**DOI:** 10.3390/cells11162491

**Published:** 2022-08-11

**Authors:** Chenglong Wang, Sabine Stöckl, Shushan Li, Marietta Herrmann, Christoph Lukas, Yvonne Reinders, Albert Sickmann, Susanne Grässel

**Affiliations:** 1Department of Orthopedic Surgery, Experimental Orthopedics, Centre for Medical Biotechnology (ZMB), University of Regensburg, 93053 Regensburg, Germany; 2IZKF Group Tissue Reg. in Musculoskeletal Dis., University Hospital & Bernhard-Heine-Centrum for Locomotion Res., University of Würzburg, 97070 Würzburg, Germany; 3Leibniz-Institut für Analytische Wissenschaften–ISAS–e.V., Bunsen-Kirchhoff-Straße 11, 44139 Dortmund, Germany; 4Medizinisches Proteom-Center, Ruhr-Universität Bochum, 44801 Bochum, Germany; 5Department of Chemistry, College of Physical Sciences, University of Aberdeen, Aberdeen AB24 3FX, UK

**Keywords:** extracellular vesicles, mesenchymal stem cells, osteogenic potential, osteogenic differentiation, adipogenic differentiation, ECM remodeling, bone regeneration

## Abstract

Osteoporosis, or steroid-induced osteonecrosis of the hip, is accompanied by increased bone marrow adipogenesis. Such a disorder of adipogenic/osteogenic differentiation, affecting bone-marrow-derived mesenchymal stem cells (BMSCs), contributes to bone loss during aging. Here, we investigated the effects of extracellular vesicles (EVs) isolated from human (h)BMSCs during different stages of osteogenic differentiation on the osteogenic and adipogenic differentiation capacity of naïve (undifferentiated) hBMSCs. We observed that all EV groups increased viability and proliferation capacity and suppressed the apoptosis of naïve hBMSCs. In particular, EVs derived from hBMSCs at late-stage osteogenic differentiation promoted the osteogenic potential of naïve hBMSCs more effectively than EVs derived from naïve hBMSCs (naïve EVs), as indicated by the increased gene expression of COL1A1 and OPN. In contrast, the adipogenic differentiation capacity of naïve hBMSCs was inhibited by treatment with EVs from osteogenic differentiated hBMSCs. Proteomic analysis revealed that osteogenic EVs and naïve EVs contained distinct protein profiles, with pro-osteogenic and anti-adipogenic proteins encapsulated in osteogenic EVs. We speculate that osteogenic EVs could serve as an intercellular communication system between bone- and bone-marrow adipose tissue, for transporting osteogenic factors and thus favoring pro-osteogenic processes. Our data may support the theory of an endocrine circuit with the skeleton functioning as a ductless gland.

## 1. Introduction

It is known that both osteoblasts and bone-marrow adipocytes derive from common precursor cells present in the bone marrow stroma, called mesenchymal stem cells [1,2,3]. Bone-marrow-derived mesenchymal stem cells (BMSCs) are able to differentiate into chondrocytes, adipocytes and osteoblasts and play a crucial role in bone repair after trauma. In addition, the cross-talk between adipose tissue and bone constitutes a homeostatic feedback loop with adipose tissue and bone regulating each other in a complex feedback system [4]. Therefore, any adipogenic or osteogenic differentiation disorder of BMSCs may contribute to bone loss during aging, and some pathological processes are accompanied by increased bone marrow adipogenesis. Accumulating evidence has clearly shown that excessive steroid application can cause a shift in the differentiation balance between osteoblasts and adipocytes, which promotes adipogenic and suppresses osteogenic differentiation of BMSCs [5], often resulting in osteonecrosis. Moreover, increasing age and osteoporotic bone disorders could facilitate adipogenic differentiation and inhibit the osteogenic differentiation of BMSCs. The accumulation of adipose tissue in the medullary cavity, which may lead to damaging the vascular endothelium, eventually results in decrease and interruption of the vascular flow in bone. Catabolic bone structural changes are likely to be the consequence [6,7,8]. Furthermore, a reduced blood supply to the bone can cause BMSCs and bone marrow to be necrotic, thereby directly affecting bone repair and reconstruction.

Extracellular vesicles (EVs) are small particles, sized 30–1000 nm, enclosed by a lipid bilayer and released by a wide range of cell types [9,10]. EVs are enriched with bioactive molecules such as lipids, proteins, mRNAs and miRNAs, and they can function as cell-to-cell communicators, as disease state transmitters, and as drug delivery vehicles in various physiological and pathophysiological processes [9,11]. In addition, the use of EVs as cell-free therapeutic tools may avoid complications such as excess immunogenicity and loss of cell viability or functionality, compared to the direct implantation of cells into the host tissues (e.g., direct implantation of MSCs into cartilage) [12,13].

Due to the promising therapeutic approach of using EVs for the treatment of non-unions after bone fracture or osteonecrosis of the femoral head, the interest in EV-based therapies in bone regeneration is increasing [14,15,16]. It has been shown that EVs derived from BMSCs can promote bone regeneration [13,14], and that there are changes in the EVs cargo and composition due to the differentiation status of the parent cells (e.g., BMSCs undergoing osteogenic differentiation) [17,18,19,20]. These studies showed that EVs isolated from BMSCs in the early or mid-stage of the osteogenic differentiation process have stronger pro-osteogenic effects than EVs from naïve BMSCs. However, none of those studies investigated the effects of EVs derived from osteogenic differentiated BMSCs on the adipogenic differentiation capacity of BMSCs. Thus, to gain a better inside into the effects of EVs derived from human (h)BMSCs during different stages of osteogenic differentiation on their pro-osteogenic differentiation capacity, our first aim was to determine the role of different EV groups (from naïve or from osteogenic differentiated hBMSCs on metabolism of naïve hBMSCs (target cells) by analyzing apoptosis, viability, and proliferation after EV treatment. Subsequently, we investigated the EV’s cargo and the effects of EVs derived from hBMSCs during different stages of osteogenic differentiation on the osteogenic and adipogenic differentiation capacity of naïve hBMSCs.

## 2. Materials and Methods

A schematic illustration of the experimental design is shown in Figure 1.

### 2.1. Ethical Statement

This study has been approved by the local ethics committee (MSCs: Ethikkommission, No. 14-101-0189, University of Regensburg, email: ethikkommission@klinik.ukr.de), and patients’ written informed consent has been obtained.

### 2.2. Isolation and Culture of Human BMSCs

hBMSCs were isolated from marrow aspirates from the femoral bone marrow cavity, obtained from ten patients undergoing hip arthroplasty surgery (mean age: 58 ± 8 years, range: 46–74 years, female: 50%). According to established protocols [21,22,23], density gradient centrifugation was used to harvest the hBMSCs. For expansion (passage 1–4), hBMSCs were cultured in StemMACS™ MSC expansion medium (Miltenyi Biotec, Bergisch Gladbach, Germany) with the following composition: L-glutamine, fetal bovine serum (FBS) [(Miltenyi Biotec, Bergisch Gladbach, Germany), and phenol red, supplemented with 0.2% MycoZap (Lonza, Basel, Switzerland)] until 80% confluency before experimental usage.

### 2.3. Osteogenic Differentiation of Human BMSCs

Before onset of osteogenic differentiation, 3 × 10^6^ hBMSCs (passage 4) were seeded in triple T175 flasks (#132867; ThermoFisher, Dreieich, Germany) and cultured in growth medium until 80% confluency. To induce osteogenic differentiation according to established protocols [23,24], the expansion medium was exchanged with osteogenic differentiation medium with the following composition: α-MEM (Sigma-Aldrich, Steinheim, Germany), 10% fetal calf serum (FCS) (Sigma-Aldrich, Steinheim, Germany) or EV-depleted FCS (FCS^depl-uc^), 1% penicillin–streptomycin (P/S) (Sigma-Aldrich, Steinheim, Germany), 4 mM GlutaMAXTM-I (Gibco, Paisley, United Kingdom), 10 µM ascorbic acid-2-phosphate (Sigma-Aldrich, Steinheim, Germany), 10 mM β-glycerophosphate (Sigma-Aldrich, Steinheim, Germany) and 100 nM dexamethasone (Sigma-Aldrich, Steinheim, Germany). Osteogenic differentiation was conducted for up to 35 days, and culture medium was replaced every 3 days.

### 2.4. Adipogenic Differentiation of Human BMSCs

Before onset of adiopogenic differentiation, 3 × 10^4^ hBMSCs (passage 4) were cultured until 100% confluency in growth medium in 6-well plates (for gene expression analysis) and in 12-well plates (for Oil red O staining). The adipogenic differentiation procedure was performed following established protocols [25]. Adipogenic differentiation medium contains DMEM high glucose (Life Technologies, Grand Island, NY, USA), 10% FCS or 10% FCS^depl-uc^, 1% P/S, 10 mg/mL insulin (Sigma-Aldrich, Saint Louis, MO, USA), 1 mM dexamethasone and 0.5 mM 3-isobutyl-1-methylxanthine (IBMX) (Sigma-Aldrich, Steinheim, Germany). Adipogenic differentiation was conducted for up to 21 days and culture medium was replaced every 3 days.

### 2.5. Generation of EVs-Depleted FCS

According to established protocols [26], FCS was diluted in α-MEM medium to a final concentration of 20%, and was ultracentrifuged at 120,000× *g* (Beckman Coulter, Brea, CA, USA) at 4 °C for 18 h. EV-depleted FCS was aliquoted and stored at −20 °C to generate EV-depleted FCS (FCS^depl-uc^) concentrates for subsequent experiments. Culture medium used for EV isolation and stimulation experiments was supplemented with 10% or 1% FCS^depl-uc^ according to experimental protocols.

### 2.6. Collection of Conditioned Medium for EV Isolation

3 × 10^6^ hBMSCs (passage 4) were seeded in triple T175 flasks in growth medium and cultured until 80% confluency. For harvesting the EV conditioned medium (CM) for EV isolation from undifferentiated naïve hBMSCs (EVs_D0 = naïve hBMSCs derived EVs), growth medium was removed and replaced by α-MEM medium with 10% FCS^depl-uc^, 1% P/S and 4 mM GlutaMAXTM-I. After 48 h incubation, CM was collected and stored immediately at −80 °C for later EV isolation. Subsequently, hBMSCs were further cultured in osteogenic differentiation medium. For harvesting osteogenic CM, the osteogenic medium was replaced by osteogenic medium with 10% FCS^depl-uc^ on days 19, 26, and 33. After 48 h incubation, the osteogenic CM was collected subsequently on days 21, 28, and 35 (i.e., EVs_D21 CM = harvesting of CM from hBMSCs undergoing osteogenic differentiation for 21 days, EVs_D28 CM = CM after 28 days, EVs_D35 CM = CM after 35 days), and stored immediately at −80 °C for EV isolation.

### 2.7. EV Isolation

Isolation of EVs from naïve hBMSCs CM and osteogenic CM was performed by ultracentrifugation as described previously [24,26,27]. In brief, the CM was centrifuged at 300× *g* (Sigma, Osterode a. Harz, Germany) and 4 °C for 10 min to remove cells, and subjected to centrifugation again at 2000× *g* to remove dead cells. Afterwards, centrifugation for 30 min at 10,000× *g* was applied to remove cell debris, then the supernatant was passed through 0.22 μm filters (Sarstedt, Nümbrecht, Germany), and was subjected to ultracentrifugation once at 120,000× *g* for 70 min at 4 °C. After careful removal of the supernatant, the pellet was resuspended with PBS and centrifuged again at 120,000× *g* for 70 min at 4 °C. The hereof resulting EV pellets were resuspended in the presence of 25 mM Trehalose (Carl Roth, Karlsruhe, Germany) in PBS. The protein concentration of EVs was measured using BCA (bicinchoninic acid) Protein Assay Kit Pierce (Thermo Scientific, Rockford, IL, USA) according to manufacturer’s instructions.

### 2.8. Western Blot Analysis for Detection of EV Markers CD9, CD63 and CD81

Protein lysates of hBMSCs were prepared by lysis of cells in RIPA lysis buffer (Carl Roth, Karlsruhe, Germany) including phosphatase (PhosphoSTOP) (Carl Roth, Karlsruhe, Germany) and proteinase inhibitors (Complete Mini) (Carl Roth, Karlsruhe, Germany). The total protein concentration in cell lysates and EV isolates was determined using a BCA assay kit (Thermo Fisher Scientific, Rockford, IL, USA). 10–30 µg hBMSC total protein lysate and 10–14 µg purified EV lysates were loaded onto and separated by 12% SDS-PAGE and transferred to 0.22 μm PVDF membranes (Carl Roche, Karlsruhe, Germany) after electrophoretic separation. The membranes were blocked with 5% BSA in Tris-buffered saline containing 0.05% Tween 20 (TBST) (0.1% Tween 20) for 1 h at room temperature and incubated with following antibodies: anti-CD9 (1:1000), anti-CD63 (1:500), and anti-CD81(1:1000) (all Thermo Fisher Scientific, Rockford, IL, USA) on a shaker overnight at 4 °C. The membranes were washed and incubated with horseradish-peroxidase-coupled secondary antibody (1:10,000) (Jackson Immuno Research, West Grove, PA, USA) for 1 h at room temperature. Protein bands were detected using SuperSignal West Femto Maximum Sensitivity Substrate (Thermo Fisher Scientific, Rockford, IL, USA) according to the manufacturers’ protocol.

### 2.9. EV Uptake Test

To investigate if EVs were incorporated by hBMSCs, similar volumes of EVs_D28 and PBS (control) were labeled with PKH26 Red Fluorescent Cell Linker Mini Kit for general cell membrane labeling (Sigma-Aldrich, Saint Louis, MO, USA) according to the manufacturer’s instructions. Afterwards, 1 × 10^4^ naïve hBMSCs were seeded in eight-well chamber slides (Falcon, Big Flats, NY, USA) and cultured in growth medium for 48 h. hBMSCs were washed with PBS and incubated with the purified PKH26-labeled EVs (10 μg/mL) and PBS in growth medium containing 10% FCS^depl-uc^ for 24 h. Nuclei of hBMSCs were counterstained with DAPI (Molecular Probes, Eugene, OR, USA) and images were captured using a fluorescence microscope (Eclipse TE2000-U; Nikon, Tokyo, Japan).

### 2.10. Nanoparticle Tracking Analysis (NTA)

The NanoSight NS300 (Malvern Instruments, Malvern, UK) was used to determine the concentration and particle size distribution of the purified EV fractions and a negative control (PBS) following the manufacturer’s instructions. Briefly, the accuracy of NTA was confirmed with 100 nm polystyrene beads (Sigma-Aldrich, Saint Louis, MO, USA), then EV samples were diluted 1:100 in PBS at room temperature and a total of five 30 s videos were recorded.

### 2.11. Proliferation Assay

The proliferation of hBMSCs stimulated with EVs was determined using a BrdU assay (Carl Roch, Mannheim, Germany). After 3000 hBMSCs were cultured in 96-well plates for 2 days in growth medium, cells were stimulated with the different EV groups (5 μg/mL) or PBS (no EVs) in growth medium containing 1% or 10% FCS^depl-uc^. Simultaneously the BrdU reagent was added according to manufacturer’s protocol. The absorbance was read at 450/690 nm with a Tecan ELISA reader (Tecan, Mannedorf, Switzerland). Results were calculated as percentage of the no EVs group (PBS).

### 2.12. Apoptosis Assay

To analyze cell apoptosis, 5000 hBMSCs were cultured in black flat-bottom 96-well plates for 2 days and treated with the different EV groups (5 μg/mL) or PBS (no EVs) in growth medium containing 1% or 10% FCS^depl-uc^. Afterward, Apo-ONE Homogeneous Caspase-3/7 assay kit (Promega Corporation, Madision, WI, USA) was used to determine the caspase 3/7-activity as indicator for apoptosis following the manufacturer’s instructions. Results were calculated as percentage of the no EVs group (PBS).

### 2.13. Viability Assay

5 × 10^3^ hBMSCs were cultured for 2 days in 96-well plates and stimulated with the different EV groups (5 μg/mL) or PBS (no EVs) in growth medium with 1% or 10% FCS^depl-uc^. The CellTiter-Blue (CTB) Cell Viability Assay (Promega, Mannheim, Germany) was used to determine cell viability according to manufacturer’s protocol. Results were calculated as percentage of the no EVs group (PBS).

### 2.14. Alkaline Phosphatase Assay

Intracellular alkaline phosphatase (ALP) enzyme activity was measured with QuantiChrom Alkaline Phosphatase Assay Kit (BioAssay Systems, Hayward, CA, USA). 1 × 10^4^ hBMSCs were seeded in 24-well plates in growth medium and cultured until 80% confluency. Growth medium was removed and replaced by osteogenic differentiation medium containing 10% FCS^depl-uc^. The medium was changed every third day. hBMSCs were kept in osteogenic differentiation medium for 14 days and were treated with the different EV groups (5 μg/mL) or PBS (no EVs) from days 7 to 14 (fresh EVs were added daily). Afterwards, intracellular ALP enzyme activity was quantified according to manufacturer’s instructions. Results were calculated as percentage of the no EVs group (PBS).

### 2.15. Alizarin Red Staining

3 × 10^4^ hBMSCs were cultured in 12-well plates in growth medium until 80% confluency. The growth medium was replaced by osteogenic differentiation medium with 10% FCS^depl-uc^ for 21 days and stimulated with the different EV groups (5 μg/mL) or PBS (no EVs) from days 6 to 21 (EVs and PBS were added daily). Cells were washed and fixed with glutaraldehyde, washed again with PBS (pH = 4.2) and incubated with Alizarin Red-S staining solution (1%, Carl Roth, Karlsruhe, Germany) for 20 min at room temperature. Images were captured using a light microscope (Eclipse TE2000-U; Nikon, Japan). Quantification of Alizarin Red staining was performed as described previously [24,28].

### 2.16. Oil Red O Staining

3 × 10^4^ hBMSCs were kept in 12-well plates in growth medium and cultured until 100% confluency. The growth medium was replaced by adipogenic medium supplemented with 10% FCS^depl-uc^ for 21 days, and the cells were treated with the different EV groups (5 μg/mL) or PBS (no EVs) from days 6 to 21 (EVs and PBS were added daily). Afterwards, cells were fixed with formaldehyde and stained with 0.2% Oil red O (Carl Roth, Karlsruhe, Germany) in 40% 2-propanol solution for 30 min at room temperature. Images were captured using a light microscope (Eclipse TE2000-U; Nikon, Japan). Subsequently, Oil Red O solution was eluted from the cells with 100% 2-propanol solution and quantified by measuring the absorbance at 492 nm as previously published [29]. Results were calculated as percentage of the no EVs group (PBS).

### 2.17. RNA Isolation and Real-Time RT-PCR

6 × 10^4^ hBMSCs were cultured in osteogenic or adipogenic differentiation medium with 10% FCS^depl-uc^ for 14 days (6-well plates) and were treated with different EV groups (5 μg/mL) or PBS (no EVs) from days 12 to 14 (EVs and PBS were added daily). Total cellular RNA from cells was isolated using the Absolutely RNA Miniprep Kit (Agilent Technologies, Cedar Creek, TX, USA) following the manufacturer’s instructions, and reverse-transcribed into cDNA using AffinityScript QPCR cDNA Synthesis Kit (Agilent Technologies, Cedar Creek, TX, USA). Real-time PCR was performed in duplicates using the Brilliant III Ultra-Fast SYBR Green QPCR Master Mix with an Agilent PCR-System (Agilent Technologies, Cedar Creek, TX, United States). All genes were analyzed relatively, normalized to GAPDH and TATA-binding protein (TBP) expression, and calibrated to the expression in the no EVs control group (calibrator). The primers are listed in Table 1.

### 2.18. Proteomics Analysis

#### 2.18.1. Sample Preparation of EVs for Proteomics Analysis

EV samples (EV_D0 and EV_D28_35) were purified from the culture supernatant of corresponding three hBMSCs culture groups, obtained from three patients, and total protein concentration was determined by BCA assay. Approximately 5 µg of total EV protein was reduced in 10 mM dithiothreitol (DTT) for 30 min at 56 °C and further alkylated in 30 mM of Iodoacetamide (IAA) for 30 min at room temperature in the dark. The remaining IAA was quenched with 30 mM dithiothreitol (DTT) for a further 15 min at room temperature. Subsequently, samples were digested using the S-Trap™ (ProtiFi, Farmingdale, NY, USA) mini procedure according to the manufacturer’s instruction.

#### 2.18.2. Quantitative Proteomic Analysis by LC-MS/MS

All peptide/protein samples were analyzed by nano LC-MS/MS using 500 ng per sample. Samples were loaded on an Ultimate 3000 Rapid Separation Liquid chromatography (RSLC) nano system with a ProFlow flow control device coupled to a Lumos Fusion orbitrap mass spectrometer (both from Thermo Scientific, Bremen, Germany). Loaded peptides were concentrated on a trapping column (Acclaim C18 PepMap100, 100 μm, 2 cm, Thermo Scientific, Bremen, Germany) using 0.1% TFA at a flowrate of 10 μL/min. For sample separation, a reversed phase column (Acclaim C18 PepMap100, 75 μm 50 cm, Thermo Scientific, Bremen, Germany) using a binary gradient was used.

#### 2.18.3. Database Search and Bioinformatics Analysis

All MS raw data were processed with Proteome Discoverer software 2.3.0.523 (Thermo Scientific, Bremen, Germany) and searched in target/decoy mode against a human Uniprot database (www.uniprot.org, downloaded 21 November 2019) using MASCOT algorithm. The search parameters were: precursor and fragment ion tolerances of 10 ppm and 0.5 Da for MS and MS/MS, respectively; trypsin set as enzyme with a maximum of 2 missed cleavages; carbamidomethylation of cysteine set as fixed modification and oxidation of methionine set as dynamic modification; using Percolator false discovery rate (strict) set to 0.01 for both peptide and protein identification. A Label-free quantification (LFQ) analysis was performed including replicates for each condition. Only proteins identified with ≥2 unique peptides, protein ratio calculation for pairwise and *t*-test (background based), *p*-value < 0.05 were considered significantly regulated. All significantly different regulated proteins were further subjected to Gene Ontology (GO) and Kyoto Encyclopedia of Genes and Genomes (KEGG) analyses. The following bioinformatics analyses including Venn diagram, GO, and KEGG et al. of annotated protein were conducted with the bioinformatics platform STRING database (http://string-db.org, accessed on 10 April 2021) and the platform Dr. Tom (https://biosys.bgi.com, accessed on 10 April 2021) provided by the Beijing Genomics Institute (BGI).

### 2.19. Statistical Analysis

Prism8.21 software (GraphPad, San Diego, CA, USA) was used for statistical analysis. Wilcoxon signed-rank test was used when the “no EVs group” was set to 100%, and two-tailed Mann–Whitney U-test or two-way ANOVA with Tukey’s multiple comparisons test was used to compare differences between the groups. *p* < 0.05 was considered statistically significant.

## 3. Results

### 3.1. EV Isolation and Characterization

Tetraspanins such as CD9, CD63 and CD81 are the most common canonical EV marker proteins, present on the vesicle surface. EVs were isolated from naïve hBMSCs CM and the different osteogenic CMs, and the presence of CD63 (Appendix A), CD9 (Appendix A) and CD81 (Appendix A) in the appropriate EV lysates was verified by Western blotting. For purification control, total BMSC lysates were also analyzed for the presence of these markers. For loading control, the respective blot membranes were stained with Ponceau Red.

### 3.2. NTA Evaluation of EVs

We determined the particle concentration and distribution of particle size of PBS (negative control), naïve EVs (EVs_D0) and osteogenic EVs (EVs_D28) via NTA (Figure 2A–C). The average particle size of both representative EV groups was about 150 nm, which corresponds to the standard size of EVs (Figure 2B,C). There is no significant difference in counts and size between EVs_D28 and EVs_D0 as calculated by NTA (Figure 2D,E).

### 3.3. Uptake of EVs by hBMSCs

PKH-26 staining of isolated EVs_D28 was performed to evaluate the internalization of EVs into hBMSCs. An intense intracellular fluorescence signal (red) was detected in hBMSC intracellular compartments (Figure 2E, merge), indicating that labeled EVs were indeed uptaken by some hBMSCs.

### 3.4. Effects of EVs Derived from Different Stages of Osteogenic Differentiating hBMSCs on Proliferation, Vitality and Apoptosis of Naïve hBMSCs

To investigate the influence of the different EV groups on the growth rate of naïve hBMSCs, cells were kept in growth medium containing 10% or 1% FCS^depl-uc^ for up to 3 days and were treated either with the different EV groups or PBS (no EVs) every day. There is no difference in cell proliferation between the five groups when cells were cultured in growth medium with 10% FCS^depl-uc^ during EV treatment from day 1 to 3 (Appendix A).

For hBMSCs cultured in medium with 1% FCS^depl-uc^, there is no difference in cell proliferation after 1 or 2 days of EV stimulation. However, after 3 days of hBMSC treatment with the EV_D21 group, a significantly enhanced proliferation of hBMSCs compared to the no EV control group, the EVs_D28 and EVs_D0 groups was detected (Appendix A). According to these data, medium containing 1% FCS^depl-uc^ and the time schedule of 3 days of EVs treatment were considered the most suitable experimental setup for proliferation assay. As shown in Figure 3A, the EVs_D0, EVs_D21, and EVs_D35 groups significantly promoted the proliferation of hBMSCs after 3 days of stimulation compared to the no EV control group.

To analyze cell viability, hBMSCs were stimulated for 3 days with EVs from osteogenic differentiated hBMSCs (EVs_D0, EVs_D21, EVs_D28, EVs_D35), or PBS (no EVs) in medium containing 10% or 1% FCS^depl-uc^. For hBMSCs cultured in medium with 10% FCS^depl-uc^, cell viability was significantly increased only after EVs_D35 treatment in comparison to no EV treatment (Appendix A).

EV stimulation of hBMSCs in medium containing 1% FCS^depl-uc^ revealed that all of the four EVs groups significantly increased the viability of hBMSCs compared to the no EVs group, and EVs_D21 and EVs_D28 groups significantly promoted cell viability compared to EVs_D35 group (Figure 3B).

For hBMSCs cultured in growth medium containing 10% FCS^depl-uc^, no differences between EV groups or when compared to the no EVs group was observed concerning apoptosis (Appendix A).

However, apoptosis was significantly decreased in hBMSC cultured in medium with 1% FCS^depl-uc^ after treatment with all four EVs groups compared to the no EVs group, whereas no difference was observed within the four different EVs groups (Figure 3C).

### 3.5. Effect of EVs Derived from Different Stages of Osteogenic Differentiated hBMSC on Osteogenic Differentiation of Naïve hBMSCs

To assess the capability of EVs from osteogenic differentiated hBMSCs to induce matrix calcification of naïve hBMSCs, quantification of calcified matrix after 21 days of osteogenic differentiation and simultaneous EV stimulation was conducted via Alizarin Red staining. Representative images show calcium deposits after Alizarin Red staining in all treated groups with a continuous increase in intensity and area comparing the no EV group to the EVs_D35 group (Figure 4A). The quantification of the Alizarin Red staining intensity confirmed these results, as all of the four experimental EV groups significantly increased calcium deposits compared to the no EV group. Notably, compared to the EVs_D0 and EVs_D21 groups, EVs_D35 significantly promoted calcium deposition. Moreover, the EVs_D28 group was more efficient than EVs_D0 (Figure 4B). We further analyzed the impact of EVs from osteogenic differentiated hBMSCs on osteogenic activity of naïve hBMSCs via ALP assay. The results demonstrated that all experimental EV groups significantly increased ALP activity compared to the no EVs group. In addition, treatment with EVs_D21 lead to a higher ALP activity level in comparison to stimulation with EVs_D0 and EVs_D35 (Figure 4C).

By analyzing osteogenic marker gene expression, we observed that OPN, BGLAP, RUNX2, and COL1A1 expression was increased after treatment with the EVs_D28 and EVs_D35 groups compared to no EVs group. Notably, compared to EVs_D0, the EVs_D28 group significantly increased gene expression levels of COL1A1 and OPN. Moreover, ALP gene expression level was significantly increased after treatment with EVs_D0, EVs_D21 and EVs_D28 compared to the no EVs group (Figure 5A–E).

### 3.6. Effects of EVs Derived from Different Stages of Osteogenic Differentiated hBMSCs on Adipogenic Differentiation of Naïve hBMSCs

Twenty-one days of adipogenic differentiation of naïve hBMSCs in the presence of EVs derived from osteogenic differentiated hBMSCs (EVs_D21, EVs_D28 and EVs_D35), for the last 15 days, resulted in a decreased number of lipid droplets compared to the no EVs group (Figure 6A). The quantification of the Oil red O staining revealed a decrease in the staining intensity for the EVs_D21, EVs_D28 and EVs_D35 groups compared to the no EVs control group (Figure 6B). Gene expression analysis of the adipogenic markers ADIPOQ, C/EBPα and PPARγ revealed that stimulation with the EVs_D35 group significantly decreased ADIPOQ expression compared to the no EV control, EVs_D21, and EVs_D28 groups. C/EBPß gene expression was reduced significantly in hBMSCs after stimulation with the EVs_D21 group compared to the no EV control, EVs_D0, and EVs_D35 groups. No differences in gene expression after EV stimulation was observed for PPARγ (Figure 6C–E).

### 3.7. Proteomic Analysis of Osteogenic EVs and Naïve-EVs

#### 3.7.1. Summary of the Proteomic Profiles

To determine the potential molecules that mediate the effects of osteogenic EVs in promoting osteogenesis and suppressing adipogenesis, proteomic analysis was performed to identify the protein profiles of EVs_28-35 and EVs_0 groups. As shown in Figure 7A, a total of 720 proteins were identified in the EVs_D28-35 group and a total of 731 proteins were identified in the EVs_D0 group. Notably, the EVs_28-35 group has 701 proteins in common with the EVs_D0 group. By setting a mean fold-change of 2 and *p* value < 0.05 to compare the EVs_D28-35 group with the EVs_D0 group, the protein levels of 175 proteins were significantly different and they were selected for further analysis. Among them, levels of 109 proteins were significantly increased and levels of 66 proteins were significantly decreased in the EVs_D28-35 group compared to the EVs_D0 group (Figure 7B). We identified 22 pro-osteogenic and 4 anti-adipogenic proteins and 8 proteins involved in ECM remodeling, which are highly enriched in EVs_D28-35 compared with that in EVs_D0 (Table 2, Table 3 and Table 4). The EVs_D28-35/EVs_D0 ratios measured for these proteins ranged from a 2.492-fold to a 100-fold increase.

#### 3.7.2. GO and KEGG Enrichment Analyses

GO analysis was performed to define all significant regulated proteins, which are split into three categories (biological processes, molecular functions and cell components). GO biological processes analysis of upregulated proteins in EVs_D28-35 compared with that in EVs_D0 matched 200 GO terms, and the six significant GO terms related to cell metabolism (proliferation, apoptosis and viability) are shown in Table 5. GO analysis determined that the upregulated cellular components in EVs_D28-35 compared with EVs_D0 include, among others, extracellular exosomes and collagen-containing extracellular matrix components. The top 20 GO terms are shown in Figure 8A. In addition, the top 20 GO terms related to molecular functions are shown in Figure 8B, which revealed that the upregulated molecular function components in EVs_D28-35 compared with EVs_D0 include, among others, extracellular matrix structural constituents, integrin binding molecules, collagen-binding- and insulin-like growth factor binding molecules. KEGG enrichment pathway analysis revealed that especially ECM–receptor interaction and the PI3K-Akt signaling pathway were significantly upregulated in EVs_D28-35 compared with EVs_D0. The top 20 upregulated pathways are shown in Figure 8C.

## 4. Discussion

In line with our previous studies [24,26,30], we characterized the EVs after their isolation from the CM via ultracentrifugation and confirmed their purity and identity. Serum deprivation (reduced serum or serum-free) often serves as a tool to study cellular viability and apoptosis and to simulate an ischemia model in vitro [31,32]. After 3 days of simultaneous stimulation with EVs and culturing with 1% FCS^depl-uc^, hBMSCs increased proliferation significantly, whereas no difference in cell proliferation was detected when cells were stimulated with the different EV groups for only 1 or 2 days. Similar to our results, Zhang et al. [33] did not observe differences in cell proliferation after the addition of 5 μg/mL MSCs-derived EVs for 1 or 2 days. Thus, we assume that the cells require sustained stimulation with EVs at least for 3 days when cultured in the presence of low serum concentrations. Serum starvation reduces basal cellular metabolism and slows down the cell cycle to achieve a proliferation-quiescent status [31]. Thus, we presumed that cells need to pass a certain threshold of stimulation time in the presence of biologically active EVs to gradually recover their cellular functions after serum starvation. However, we did not observe a difference in cell proliferation when cells were kept in medium with 10% FCS^depl-uc^ during treatment with the different EV groups, suggesting that the effects of EVs are masked by a too-high FCS components concentration. Our results are consistent with the study of Qin et al. [34], showing that hBMSCs-derived EVs do not have any effects on cell proliferation during EVs treatment from day 1 to 7 when cells were cultured with 10% FCS. We conclude that EVs can promote proliferation and other metabolic activities significantly only when cells are cultured in a growth-factor-starved microenvironment (e.g., 1% or less FCS).

EVs from both naïve and osteogenic differentiated hBMSCs significantly increased the proliferation and viability of naïve hBMSCs, and decreased apoptosis, compared to the control group that received no EV treatment, underlining the pro-anabolic influence of EVs in general. However, we did not observe a difference between EVs derived from osteogenic differentiated and naïve hBMSCs concerning the proliferation, apoptosis and viability of the target cells. The proteomic profiling of EVs and subsequent GO-enriched analysis showed that a similar number of proteins are upregulated both in osteogenic EVs and naïve EVs, which positively (22) and negatively (13) influence the regulation of cell proliferation (Table 5) promoting similar cell responses. Similar effects may apply for viability and apoptosis. It is known that the catalytic activity of a variety of enzymes is essential to maintain cell viability [35]. We identified 21 proteins involved in “positive regulation of catalytic activity” and 15 proteins involved in “negative regulation of catalytic activity”, which are highly enriched in osteogenic EVs compared to naïve EVs (Table 5). Although the osteogenic differentiation state of the parent cells affects EVs cargo, this does not change cell proliferation, apoptosis and viability in our study, indicating that basic metabolic activities are independent of the differentiation state.

hBMSCs-derived EVs, harvested at various stages along the osteogenic differentiation pathway of the parent cells can promote the osteogenic differentiation capacity of naïve hBMSCs characterized by increased calcium deposits, enhanced ALP activity and by the induction of expression of the osteogenic marker genes OPN, BGLAP, RUNX2, ALP, and COL1A1. Our proteomic analysis data support those results, showing that several pro-osteogenic proteins such as fibronectin (FN1) and cartilage oligomeric matrix protein (COMP) were highly concentrated both in EVs_D0 and EVs_D28-35, with COMP being more concentrated in the EVs_D28_35. FN 1 is a macromolecular component of the extrafibrillar ECM, which promotes the osteogenic differentiation of stem cells by inducing the gene expression of osteogenic markers [36]. COMP, a critical component of the collagen fibrillar network, can contribute to enhanced osteogenesis by inducing ALP activity, matrix mineralization, and the expression of such genes as BGLAP, RUNX2, and ALP [37]. In particular, EVs derived from hBMSCs from late stages of osteogenic differentiation (EVs_D28, EVs_D35) support the osteogenic capacity of hBMSCs stronger compared to naïve EVs. Our results are consistent with the study of Narayanan et al. [20], showing that EVs from hBMSCs undergoing osteogenic differentiation for 28 days can promote bone regeneration, both in vitro and in vivo, and were more efficient in inducing osteogenesis of the hBMSCs judged by increased matrix mineralization in vivo. A higher mineral content in the EVs from late-stage osteogenic differentiated BMSC might be critical for their higher osteogenic inducing potential.

Increased ALP activity is characteristic for committed pre-osteoblasts in an early stage of osteogenic differentiation, whereas in the mature osteoblasts ALP activity is reduced [38]. We suggest that EVs can transmit the differentiation state of the secreting parent cells to the target cells supported by our observation that EVs_D21 exhibited a stronger ability to enhance ALP activity compared to EVs from later differentiation stages [39]. Together with the results of the viability assay, the strongest effects were induced from the EVs_D21 group, suggesting that cell viability might decrease in the late phase of osteogenic differentiation. Our proteomic data support these data revealing that osteogenic differentiation affects EVs’ cargos. Twenty-two pro-osteogenic proteins were more concentrated in osteogenic EVs compared to native EVs (Table 2). In particular, Fibulin-1 (FBLN1), Prolargin (PRELP) and Matrix Gla protein (MGP) concentration is increased from 63.867-fold to 100-fold in EVs_D28_35 compared to naive EVs. FBLN1 was recently described as a new positive modulator of bone repair and regeneration [40]. PRELP as a collagen-binding proteoglycan, which is highly expressed in the developing bone can inhibit osteoclastogenesis and modulate osteoblast differentiation [41]. MGP can promote osteoblast differentiation and mineralization [42]. Furthermore, our GO analysis concerning molecular functions and cell components revealed upregulation of proteins involved in ECM homeostasis, ECM-related protein complexes and binding (fibronectin, a variety of laminins and collagens) [43], and focal adhesion. In bone, the ECM is an intricate dynamic microenvironment that plays an essential role by instructing the differentiation process of BMSCs to osteoblasts and by favoring the absorption of bone by osteoclasts [44]. Focal adhesion plaques are specialized structures formed at the cell–ECM contact points, which critically regulate intracellular signaling after ECM–receptor interaction [45].

In addition, KEGG enrichment analysis identified upregulated intracellular signaling pathway components related to bone regeneration in the osteogenic EVs compared to native EVs. Therefore, ECM–receptor interaction and PI3K/Akt signaling pathway components revealed the strongest regulation, and activation of the PI3K/Akt pathway has been shown to play a key role in osteoblast proliferation, differentiation and bone formation [46]. This might be one of the underlying molecular reasons why EVs_D28-35 are more efficient in inducing osteogenesis. Cell surface receptors such as integrin receptors are critical in mediating the transmission of external stimuli into the cell, and ECM substrates can promote osteogenic differentiation of MSCs through ECM–receptor interaction (e.g., via integrin receptors) [45].

We observed that all EV groups except the one from naïve hBMSCs suppressed the adipogenic differentiation capacity of hBMSCs, indicated by a decrease in the gene expression of pro-adipogenic genes such as C/EBPβ and ADIPOQ. C/EBPβ is a transcription factor, which plays a critical role in adipogenesis [47], and adiponectin is an adipokine, specifically secreted by adipocytes [48]. Only a few studies have focused on the effects of EVs on the adipogenic differentiation capacity of BMSCs, but a recent publication [15] reported that “healthy” rat BMSCs-derived EVs can suppress the adipogenic differentiation capacity of BMSCs isolated from rats after steroid-induced femoral head necrosis (SFHN) compared to no EVs treatment. In contrast to those data, in our study, naïve EVs were not able to inhibit adipogenesis. A possible reason for that might be the fact that the hBMSCs that we used are from osteoarthritic patients instead of SFHN patients.

In 2007, Lee et al. [49] argued for the first time that there might exist an endocrine circuit with the skeleton functioning as a ductless gland, with osteoblasts secreting the pro-osteogenic hormone osteocalcin that profoundly influences energy metabolism. Our study supports this theory, as our results suggest osteogenic EVs exert negative effects on the adipogenic differentiation capacity of naïve hBMSCs. We suggest that four anti-adipogenic proteins, which are highly enriched in osteogenic EVs, AEBP1, CCDC80, APOD and PLSCR3 (Table 3), might promote these anti-adipogenic effects. AEBP1 is down-regulated during adipogenesis, and negatively regulates the differentiation of pre-adipocytes [50]. CCDC80 is a secreted protein and its overexpression can inhibit adipogenesis [51]. Interestingly, CCDC80 is also highly enriched in human white adipose tissue [51]. APOD is an atypical apolipoprotein, which contributes to enhanced triglyceride metabolism and plays a significant role in lipid homeostasis [52]. PLSCR3 is a novel candidate gene capable of influencing adipose cell function and is decreased during adipogenic differentiation. Overexpression of human PLSCR3 can suppress triacylglycerol accumulation and expression of pro-adipogenic transcription factors [53].

Thus, we speculate that osteogenic EVs could serve as an intercellular communication system between bone and progenitor cells for transporting factors affecting lipid homeostasis, which in turn affect bone marrow adipose tissue formation negatively (Figure 9).

Finally, bone loss and reduced bone formation is accompanied by increased marrow adiposity in osteoporosis. This is a major clinical sequela for patients with malignant diseases (e.g., lymphoma, leukaemia) undergoing chemotherapy, radiation therapy or recovering from osteoporotic fractures [54]. Therefore, if an agent or drug could alter the cellular lineage allocation early in the pathologic bone marrow to favor osteogenesis and suppress adipogenesis from precursor MSC [55], this could have a major inhibiting effect on osteoporosis morbidity and tremendous cost savings.

## 5. Conclusions

In conclusion, the present study shows that EVs, both from naïve and osteogenic differentiated hBMSCs, do improve the viability, and reduce the apoptosis, of naïve hBMSCs. EVs derived from hBMSCs in the late stage of osteogenic differentiation promote the osteogenic differentiation potential of naïve hBMSCs to a greater extent than EVs derived from naïve hBMSCs. Both EV groups have distinct protein profiles, with pro-osteogenic and anti-adipogenic proteins encapsulated preferentially in osteogenic EVs. EVs treatment might, therefore, be a promising non-cellular and non-drug-mediated approach to positively influence bone health. Our data speculate that osteogenic EVs exert positive effects on the osteogenic differentiation capacity of precursor cells, and negative effects on the adipogenic differentiation capacity of hBMSCs. We hypothesize that these EVs can serve as an intercellular communication system between bone tissue and progenitor cells for transporting osteogenic hormones/factors, which in turn affect bone marrow adipose tissue formation negatively. This interplay supports the theory of an endocrine circuit within the skeleton functioning as a ductless gland.

## Figures and Tables

**Figure 1 cells-11-02491-f001:**
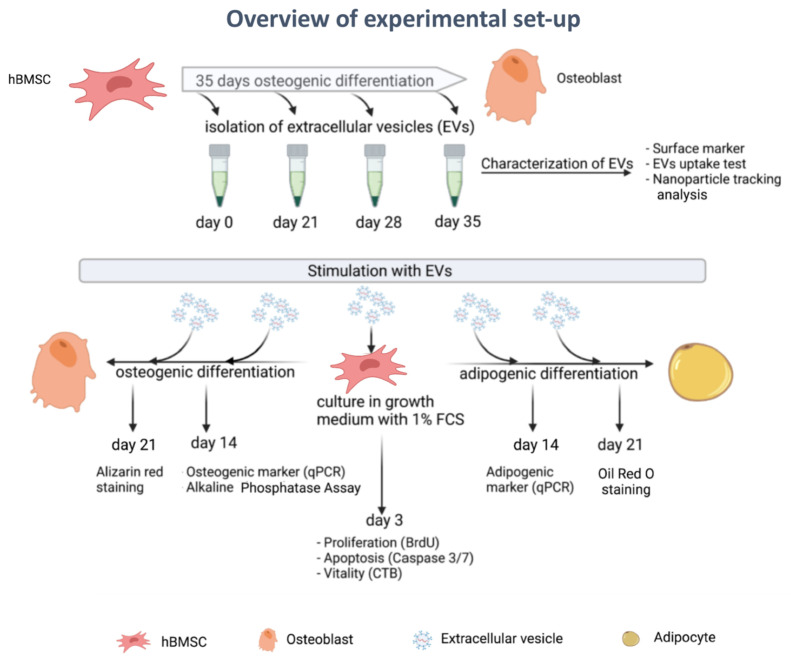
Overview of experimental set-up; EVs = extracellular vesicles.

**Figure 2 cells-11-02491-f002:**
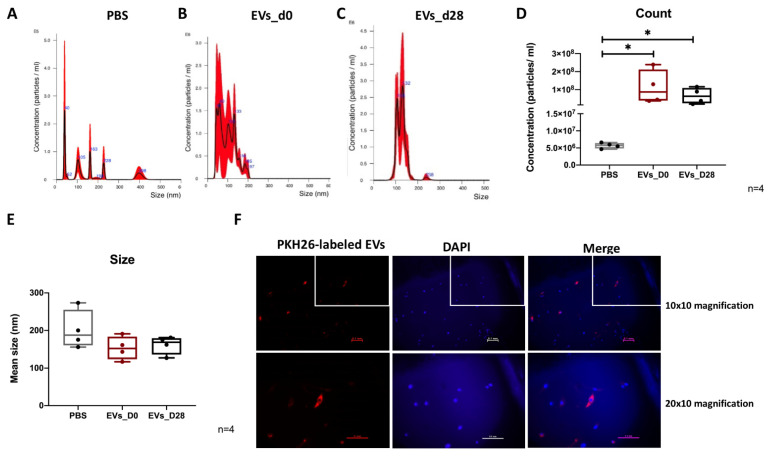
**Molecular characterization of EVs.** (**A**–**C**) Particle size distribution of PBS, naïve hBMSCs-EVs and osteogenic EVs was measured by NTA analysis. n = 4. (**D**,**E**) Quantitative comparison among PBS, naïve hBMSCs-EVs and osteogenic EVs in count and size measured by NTA analysis. n = 4. * *p* < 0.05; two-tailed Mann–Whitney U-test was used to compare differences between different groups. (**F**) Uptake of EVs by hBMSCs. PKH26-labeled EVs_D28 (red) were internalized by hBMSCs and visualized with fluorescence microscopy. Cell nuclei were stained with DAPI. Scale bar: 100 μm. n = 2.

**Figure 3 cells-11-02491-f003:**
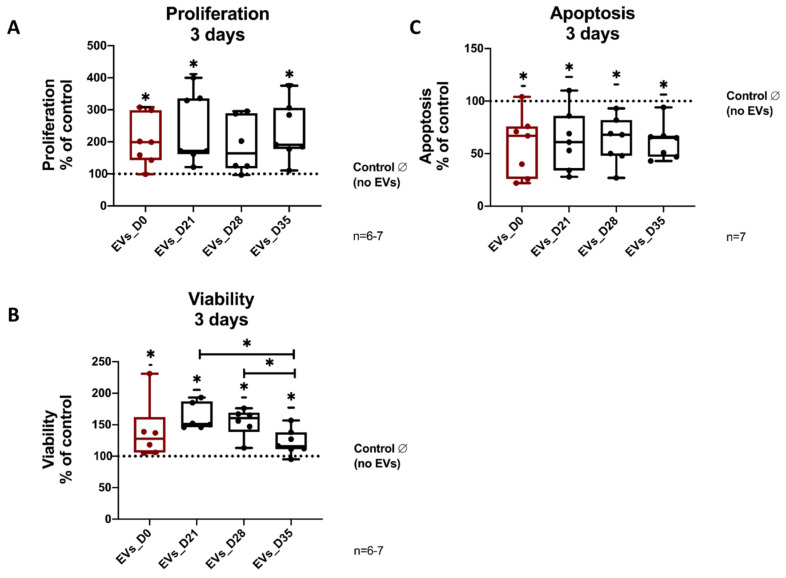
**Effects of EVs derived from different stages of BMSC osteogenesis on naïve hBMSC proliferation, viability and apoptosis.** (**A**) For proliferation analysis, naïve hBMSCs were stimulated for 3 days with the different EV groups or PBS (no EVs) in medium containing 1% EV-depleted FCS; n = 6–7. Analysis of viability (**B**) and apoptosis (**C**) of naïve hBMSCs after stimulation for 3 days with the different EV groups or PBS (no EVs) in medium containing 1% EVs-depleted FCS; n = 6–7. Results were calculated as percentage to the control group (no EVs, shown by the dotted line); * *p* < 0.05; Wilcoxon signed-rank test was used when no EVs group was set to 100%, and two-tailed Mann–Whitney U-test was used to compare differences between different EV-treated groups.

**Figure 4 cells-11-02491-f004:**
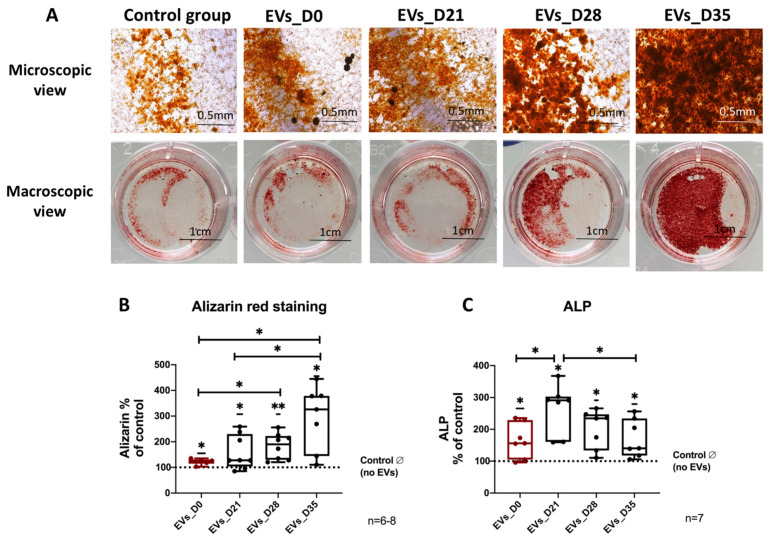
**Evaluation of osteogenic differentiation of naïve hBMSCs after EV treatment.** (**A**) Alizarin Red staining of hBMSCs after 21 days of osteogenic differentiation and treatment with the different EVs groups. Microscopic view (scale bar: 0.5 mm) and macroscopic view (scale bar: 1 cm); n = 6–8. (**B**) Quantification of Alizarin Red staining; n = 6–8. (**C**) Alkaline Phosphatase (ALP) activity of hBMSC after 14 days of osteogenic differentiation and treatment with the different EV groups; n = 7. Results were calculated as percentage to the control group (no EVs, shown by the dotted line); * *p* < 0.05, ** *p* < 0.01; Wilcoxon signed-rank test was used when no EVs group was set to 100%, and two-tailed Mann–Whitney U-test was used to compare differences between different EV-treated groups.

**Figure 5 cells-11-02491-f005:**
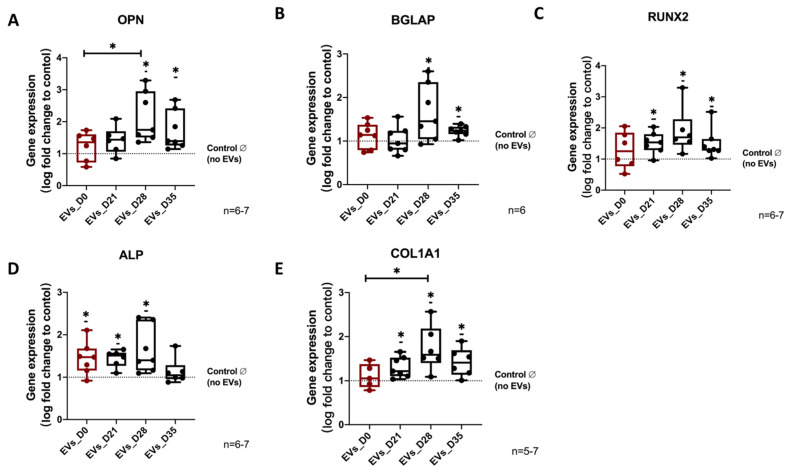
**Evaluation of osteogenic marker gene expression in hBMSCs after EV treatment.** (**A**–**E**) Gene expression level of the osteogenic marker genes (OPN, BGLAP, ALP, RUNX2, and COL1A1) were analyzed after 14 days of osteogenic differentiation of hBMSCs and simultaneous stimulation with the different EV groups. Results were calculated as percentage to the control group (no EVs, shown by the dotted line); * *p* < 0,05; Wilcoxon signed-rank test was used when no EVs group was set to 100%, and two-tailed Mann–Whitney U-test was used to compare differences between different EV-treated groups; n = 5–7.

**Figure 6 cells-11-02491-f006:**
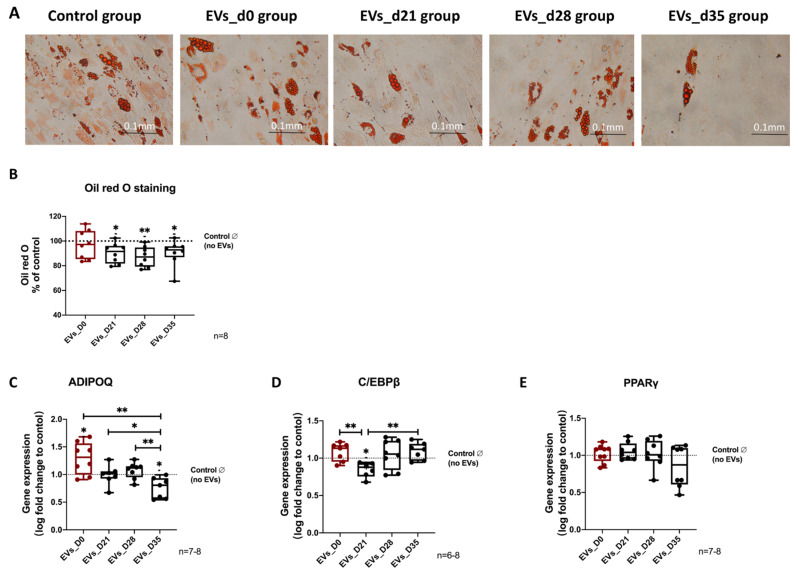
**Evaluation of adipogenic differentiation of naïve hBMSCs after EV treatment.** (**A**) Microscopic view of Oil red O staining of hBMSCs after 21 days of adipogenic differentiation of hBMSCs and treatment with the different EVs groups; n = 8. (**B**) Quantification of Oil red O staining; n = 8. (**C**–**E**) Gene expression level of adipogenic markers such as ADIPOQ, C/EBPα, and PPARγ after 14 days of adipogenic differentiation and simultaneous stimulation with the different EVs groups; n = 6–8. Results were calculated as percentage to the control group (no EVs, shown by the dotted line); * *p* < 0.05, ** *p* < 0.01; Wilcoxon signed-rank test was used when no EVs group was set to 100%, and two-tailed Mann–Whitney U-test was used to compare differences between different EV-treated groups. Scale bar: 100 μm.

**Figure 7 cells-11-02491-f007:**
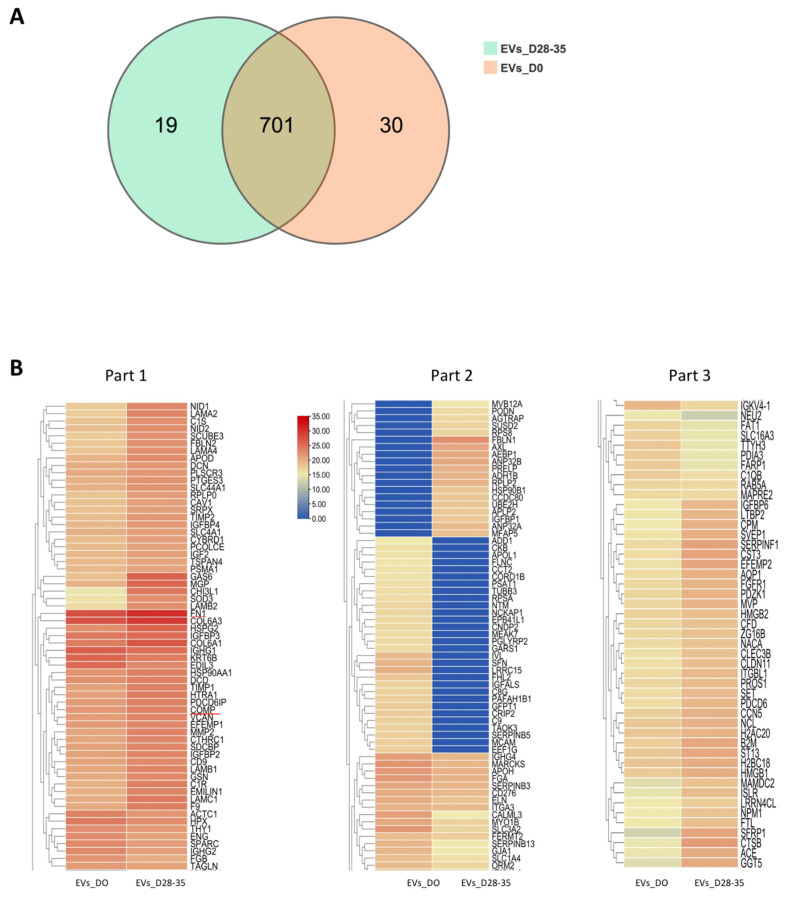
**Venn diagram of total proteins and heatmap of significantly different proteins.** (**A**) Venn diagram showing the distinct profiles of total proteins in the EVs_D28-35 and EVs_D0; n = 3. (**B**) Heatmap showing the distinct significant protein profiles of the EVs_D28-35 compared to EVs_D0; n = 3.

**Figure 8 cells-11-02491-f008:**
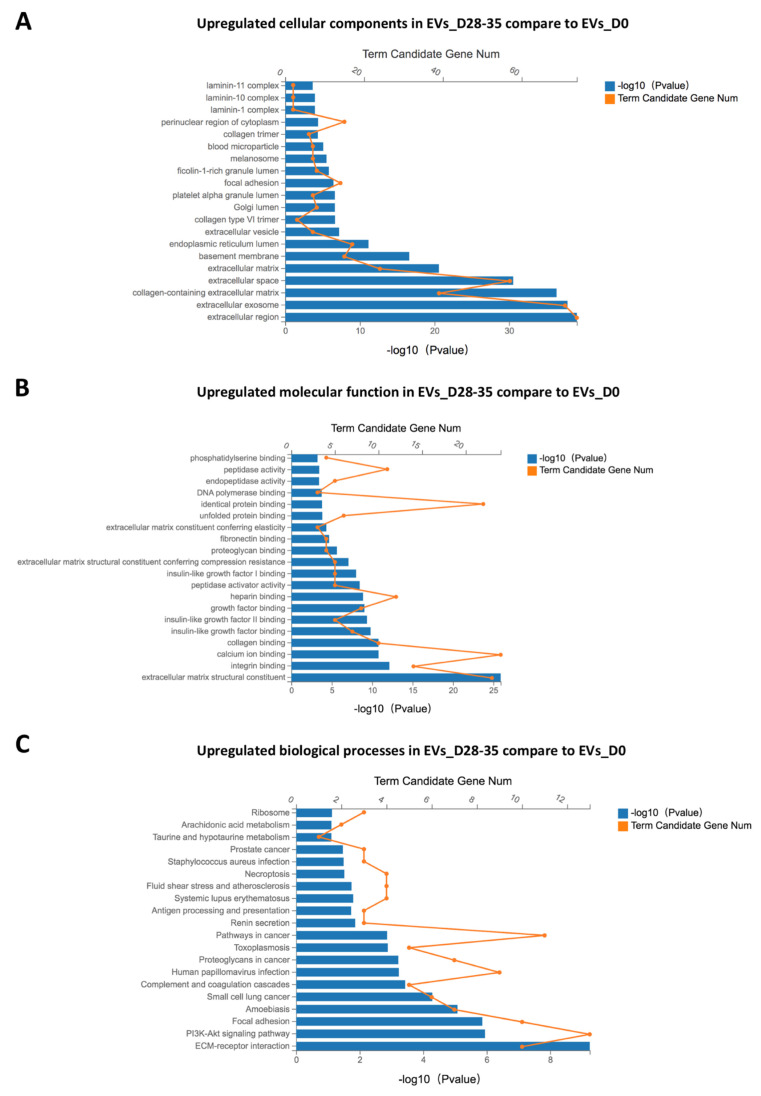
**Functional enrichment analysis of significantly regulated proteins.** Gene ontology (GO) analysis for the top 20 terms of upregulated cellular components (**A**), molecular functions (**B**), and biological processes (**C**). Kyoto Encyclopedia of Genes and Genomes enrichment analysis for significant proteins were clustered, and the top 20 pathways are shown; n = 3. Note: −log10 (*p* = 0.05) = 1.30 (the bottom horizontal axis number). −log10 (*p* = 0.01) = 2. −log10 (*p* = 0.001) = 3.

**Figure 9 cells-11-02491-f009:**
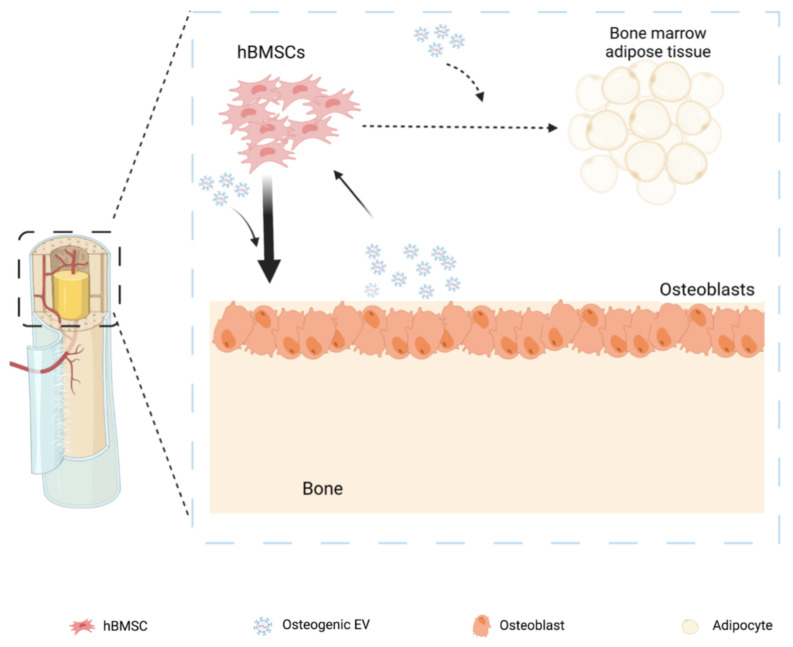
Summary Osteogenic primed EVs positively regulate osteogenic differentiation and negatively regulate adipogenic differentiation capacity of naïve hBMSCs.

**Table 1 cells-11-02491-t001:** Primer sequences for qPCR.

Gene	Primer Sequences (5′–3′)
GAPDH	Fwd, CTGACTTCAACAGCGACACC
	Rev, CCCTGTTGCTGTAGCCAAAT
TBP	Fwd, TTGTACCGCAGCTGCAAAAT
	Rev, TATATTCGGCGTTTCGGGCA
ALP	Fwd, CCTCCTCGGAAGACACTCTG
	Rev, CCTCCTCGGAAGACACTCTG
BGLAP	Fwd, GTGCAGAGTCCAGCAAAGGT
	Rev, TCAGCCAACTCGTCACAGTC
COL1A1	Fwd, ACGTCCTGGTGAAGTTGGTC
	Rev, ACCAGGGAAGCCTCTCTCTC
RUNX2	Fwd, CGGAATGCCTCTGCTGTTATG
	Rev, GCTTCTGTCTGTGCCTTCTG
OPN	Fwd, TGAAACGAGTCAGCTGGATG
	Rev, TGAAATTCATGGCTGTGGAA
C/EBPβ	Fwd, AACCTGGAGACGCAGCACAA
	Rev, GAACAAGTTCCGCAGGGTGC
PPARγ	Fwd, GACCAAAGCAAAGGCGAGGG
	Rev, CCCTGAAAGATGCGGATGGC
ADIPOQ	Fwd, AAGGAGATCCAGGTCTTATTGGTC
	Rev, CGAATGGGCATGTTGGGGAT

**Table 2 cells-11-02491-t002:** The expression/level ratio of selected pro-osteogenic proteins in EVs_D28-35 compared with that in EVs_D0.

UniProt	Protein Names	Gene Names	EVs_D28-35 /EVs_D0 Ratio
P23142	Fibulin-1	FBLN1	100
P51888	Prolargin	PRELP	100
P08493	Matrix Gla protein	MGP	63.867
P01034	Cystatin-C	CST3	7.567
Q8IX30	Signal peptide, CUB and EGF-like domain-containing protein 3	SCUBE3	7.149
P11362	Fibroblast growth factor receptor 1	FGFR1	6.795
O76076	WNT1-inducible-signaling pathway protein 2	CCN5	6.312
P02751	IGFBP4	FN1	6.29
Q96CG8	Collagen triple helix repeat-containing protein 1	CTHRC1	5.782
P22692	Insulin-like growth factor-binding protein 4	IGFBP4	4.895
P24592	Insulin-like growth factor-binding protein 6	IGFBP6	4.756
P07585	Decorin	DCN	4.365
P05452	Tetranectin	CLEC3B	4.298
P08253	72 kDa type IV collagenase	MMP2	4.06
P01344	Insulin-like growth factor II	IGF2	4.017
P18065	Insulin-like growth factor-binding protein 2	IGFBP2	3.611
P17936	Insulin-like growth factor-binding protein 3	IGFBP3	3.582
P09429	High mobility group protein B1	HMGB1	3.223
P12109	Collagen alpha-1(VI) chain	COL6A1	3.163
P13611	Versican core protein	VCAN	3.093
P49747	Cartilage oligomeric matrix protein	COMP	2.569
Q4LDE5	Sushi, von Willebrand factor type A, EGF and pentraxin domain-containing protein 1	SVEP1	2.515

**Table 3 cells-11-02491-t003:** The expression/level ratio of selected anti-adipogenic proteins in EVs_D28-35 compared with that in EVs_D0.

UniProt	Protein Names	Gene Names	EVs_D28-35 /EVs_D0 Ratio
Q8IUX7	Adipocyte enhancer-binding protein 1	AEBP1	100
Q76M96	Coiled-coil domain-containing protein 80	CCDC80	100
P05090	Apolipoprotein D	APOD	8.375
Q9NRY6	Phospholipid scramblase 3	PLSCR3	2.635

**Table 4 cells-11-02491-t004:** The expression/level ratio of selected proteins involved in ECM remodeling in EVs_D28-35 compared with that in EVs_D0.

UniProt	Protein Names	Gene Names	EVs_D28-35 /EVs_D0 Ratio
P55268	Laminin subunit beta-2	LAMB2	6.053
Q9Y6C2	EMILIN-1	EMILIN1	5.021
P07942	Laminin subunit beta-1	LAMB1	4.939
P98095	Fibulin-2	FBLN2	4.838
Q14112	Nidogen-2	NID2	3.652
Q16363	Laminin subunit alpha-4	LAMA4	3.344
Q15113	ProcollagenC-endopeptidase enhancer 1	PCOLCE	2.897
P12111	Collagen alpha-3(VI) chain	COL6A3	2.492

**Table 5 cells-11-02491-t005:** GO terms related to cell metabolism (proliferation, apoptosis, viability) in the GO biological process analysis of upregulated proteins in EVs_D28-35 compared with that in EVs_D0.

GO-Term ID	Term Description	Observed Gene Count	False Discovery Rate (*p* Value)	Candidate Gene Name
GO:0008284	Positive regulation of cell population proliferation	22	8.71 × 10^−6^	TIMP1, MMP2, SFRP1, COMP, LAMB1, IGFBP2, LAMC1, SDCBP, PDCD6, HMGB2, NPM1, AQP1, CTHRC1, FBLN1, GAS6, HMGB1, FN1, HTRA1, CST3, IGF2, FGFR1, NACA
GO:0008285	Negative regulation of cell population proliferation	13	0.0125	SFRP1, SERPINF1, TIMP2, NPM1, IGFBP6, PODN, FBLN1, CAV1, APOD, SRPX, IGFBP3, CD9, B2M
GO:0060548	Negative regulation of cell death	17	0.0047	TIMP1, SFRP1, COMP, SERPINF1, HMGB2, NPM1, HSP90B1, AXL, AQP1, GAS6, CAV1, FN1, SET, WISP2, HSPG2, CST3, NACA
GO:0031324	Negative regulation of cellular metabolic processes	28	0.0293	TIMP1, SFRP1, AEBP1, SERPINF1, SDCBP, SLC4A1, TIMP2, APLP2, COL6A3, HMGB2, NPM1, AQP1, NCL, FBLN1, GAS6, HIST2H2AC, CAV1, APOD, HMGB1, SET, EMILIN1, IGFBP3, PROS1, MVP, CST3, IGF2, FGFR1, NACA
GO:0043085	Positive regulation of catalytic activity	21	0.0070	SFRP1, PCOLCE, CHI3L1, TIMP2, PDCD6, ACE, HMGB2, AXL, IGFBP6, FBLN1, GAS6, HSP90AA1, CAV1, HMGB1, ANP32B, FN1, GSN, IGFBP3, IGF2, FGFR1, PTGES3
GO:0043086	Negative regulation of catalytic activity	15	0.0051	TIMP1, SFRP1, SERPINF1, SLC4A1, TIMP2, APLP2, COL6A3, NPM1, AQP1, GAS6, CAV1, SET, PROS1, MVP, CST3

## Data Availability

The mass spectrometry proteomics data have been deposited into the ProteomeXchange Consortium via the PRIDE partner repository with the dataset identifier PXD035845.

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
