# Peer review of "Effects of Extracellular Vesicles from Osteogenic Differentiated Human BMSCs on Osteogenic and Adipogenic Differentiation Capacity of Naïve Human BMSCs"

_cells, 2022, doi:10.3390/cells11162491_

Round 1

Reviewer 1 Report

The current manuscript aims to study the effects of EVs isolated from human BMSCs during different stages of osteogenic differentiation on osteogenic and adipogenic differentiation capacity of naïve (undifferentiated) hBMSCs.

The experiments were well designed, Materials and methods parts contain detailed protocol. The results and discussion are well written and the conclusion is based on their findings. 

The manuscript can be accepted after minor revision.

Results
3.1. EV isolation and characterization:  in Figure S1 the authors showed results obtained from D21. Did the authors characterize EVs at 28 and 35 days?

3.3. Uptake of EVs by hBMSCs

The authors showed the uptake only of EVs_D28: did was evaluated the uptake of EVs_D0? Were any differences observed? is it possible to associate a quantification of the uptake?

Author Response

Reviewer 1

We are grateful for your opinion on the manuscript and your helpful comments. Thanks to your comments we were able to improve the quality and impact of our manuscript. In the following you will find our point-by-point responses to each of the comments of yours. 

Comment: EV isolation and characterization: in Figure S1 the authors showed results obtained from D21. Did the authors characterize EVs at 28 and 35 days?

Response: Yes, we did. We have now provided Western Blot images for the classical surface markers (CD81, 63 and 9) of EVs at isolation days 28 and 35. The new images are integrated in Suppl. figure 1.

Comment: The authors showed the uptake only of EVs_D28: did was evaluated the uptake of EVs_D0?

Response: Yes, we also have evaluated the uptake of EVs_D0. Please see the figure documenting the uptake of EVs_D0 for your information (attached file).

Comment: Were any differences observed? is it possible to associate a quantification of the uptake?

Response:  There is no obvious difference in up take ratio or manner between the EV batches from the 2 different isolation days, all EVs seemed to be uptaken by the hBMSCs in the same manner. It is not possible to quantify the uptake. However, for your information, our group has published several papers1,2 about EVs, such as curcumin-primed human BMSC-derived EVs, which all show a similar uptake route.

  1. Li S, Stöckl S, Lukas C, Herrmann M, Brochhausen C, König MA, Johnstone B, Grässel S (2021) Curcumin primed human BMSC derived extracellular vesicles reverse IL-1β induced catabolic responses of OA chondrocytes by upregulating miR-126-3p. Stem Cell Research & Therapy 12:252 https://doi.org/10.1186/s13287-021-02317-6
  2. Niedermair T, Lukas C, Li S, Stöckl S, Craiovan B, Brochhausen C, Federlin M, Herrmann M and Grässel S (2020) Influence of Extracellular Vesicles Isolated From Osteoblasts of Patients With Cox-Arthrosis and/or Osteoporosis on Metabolism and Osteogenic Differentiation of BMSCs. Front. Bioeng. Biotechnol. 8:615520. doi: 10.3389/fbioe.2020.615520

Reviewer 2 Report

In their manuscript, the authors examine the effects of extracellular vesicles derived from human BMSC at several time points of osteogenic differentiation on osteogenic and adipogenic differentiation of BMSC. Moreover, they perform proteomic analysis of these EVs.

The manuscript is well written, materials and methods are sufficient to reproduce the experiments.

Minor comments

- please describe EV isolation procedure briefly. Please describe quantification of the harvested EVs.

- can the authors rule out that the osteogenic diffentiation factors added to the medium (ascorbic acid, glycerophosphate, dexamethasone) are also encapsulated in the vesicles and so promote osteogenesis in the target cells? Please discuss.

- considering this being an in vitro study, conlusions drawn on potential in vivo implications strictly must be flagged as speculative throughout the manuscript.

Author Response

We are grateful for your opinion on the manuscript and your helpful comments. Thanks to your comments we were able to improve the quality and impact of our manuscript. In the following you will find our point-by-point responses to each of the comments of yours.

Reviewer 2

Comment: please describe EV isolation procedure briefly. Please describe quantification of the harvested EVs.

Response: As suggested, we have provided a brief description of the EV isolation procedure and the quantification of the harvested EVs. This can be found at page 4 under Materials & Methods section: “2.7. EV isolation”.

Comment: can the authors rule out that the osteogenic diffentiation factors added to the medium (ascorbic acid, glycerophosphate, dexamethasone) are also encapsulated in the vesicles and so promote osteogenesis in the target cells? Please discuss.

Response: This is an important comment. Firstly, we cannot exclude that osteogenic medium components were encapsulated in the osteogenic EVs (EVs_D21, EVs_D28 and EVs_D35), but compared to EVs_D21, EVs_D35 significantly promoted calcium deposition (as documented by Alizarin Red staining) and decreased ALP activity (Fig. 4). Thus, if osteogenic medium components are encapsulated in the EVs, those are not the main ingredients for promoting osteogenesis in the target cells, otherwise there would be no difference of the effects between EVs_D21 and EVs_D35. Similarly, there is not difference between naïve hBMSC derived EVs and EVs_D35 in the ALP assay (Fig.4C). If osteogenic medium components (ascorbic acid, glycerophosphate, dexamethasone) have an effect on the osteogenic cargo of EVs, EVs_D35 should enhance ALP activity compared to EVs_D0.

Secondly, in all osteogenic assays (Alizarin Red staining, ALP assay, qRT-PCR of osteogenic markers), hBMSCs were always kept in osteogenic differentiation medium when they were treated with the different EV groups or PBS (no EVs), so each group is already exposed to osteogenic differentiation factors. Maybe with the EVs_D35 group, the target cells were exposed to a higher dexamethasone concentration, but high concentrations of dexamethasone will promote adipogenic differentiation instead of osteogenic differentiation, which we did not observe. In this line, the papers listed below did use a similar protocol for stimulation of osteogenic differentiation of MSC with osteogenic EVs.

  1. Narayanan R, Huang CC, Ravindran S. Hijacking the Cellular Mail: Exosome Mediated Differentiation of Mesenchymal Stem Cells.Stem Cells Int. 2016; 2016:3808674. doi:10.1155/2016/3808674
  2. Wang X, Omar O, Vazirisani F, Thomsen P, Ekström K. Mesenchymal stem cell-derived exosomes have altered microRNA profiles and induce osteogenic differentiation depending on the stage of differentiation.PLoS One. 2018; 13(2):e0193059. Published 2018 Feb 15. doi:10.1371/journal.pone.0193059

Comment: considering this being an in vitro study, conclusions drawn on potential in vivo implications strictly must be flagged as speculative throughout the manuscript.

Response: As suggested, the wording in abstract and conclusions has been modified appropriately.